# Genetic network shaping Kenyon cell identity and function in *Drosophila* mushroom bodies

Pei-Chi Chung, Kai-Yuan Ku, Sao-Yu Chu, Chen Chen, Hung-Hsiang Yu*

Institute of Cellular and Organismic Biology, Academia Sinica, Taipei, Taiwan

## eLife Assessment

This **fundamental** study uses the Drosophila mushroom body as a model to understand the molecular machinery that controls the temporal specification of neuronal cell types. With **convincing** experimental evidence, the authors make the finding that the Pipsqueak domain-containing transcription factor Eip93F plays a central role in specifying a later-born neuronal subtype while repressing gene expression programs for earlier subtypes.

*For correspondence:
samhhyu@gate.sinica.edu.tw

**Competing interest:** The authors declare that no competing interests exist.

**Abstract** Revealing the molecular mechanisms underlying neuronal specification and acquisition of specific functions is key to understanding how the nervous system is constructed. In the *Drosophila* brain, Kenyon cells (KCs) are sequentially generated to assemble the backbone of the mushroom body (MB). Broad-complex, tramtrack, and bric-à-brac zinc finger transcription factors (BTBzf TFs) specify early-born KCs, whereas the essential TFs for specifying late-born KCs remain unidentified. Here, we report that Pipsqueak domain-containing TF *Eip93F* promotes the identity of late-born KCs by reciprocally regulating gene expression in main KC types. Moreover, *Eip93F* not only regulates the expression of calcium channel *Ca-α1T* in late-born KCs to functionally control animal behavior, but it also forms a genetic network with *BTBzf TFs* to specify the identities of main KC types. Our study provides crucial information linking KC-type diversification to unique function acquisition in the adult MB.

## Introduction

The nervous system contains a diverse ensemble of neuron types, which are generated during development in appropriate numbers under tightly controlled processes. A central question in developmental neurobiology concerns how diverse neuron types acquire unique cell identities with characteristic morphologies, distinct gene expression profiles, and specific functionalities. Kenyon cells (KCs) are the intrinsic neurons in the *Drosophila* mushroom body (MB) (*Lee et al., 1999*) and can serve as an excellent model system for investigations into the molecular mechanisms of neuron-type diversification. Around 2000 KCs generated by four neuroblasts are grouped into three sequentially born types, including γ, α'/β' and α/β neurons (*Lee et al., 1999*). Functionally, these main KC types play different roles in short-term memory, acquisition, stabilization, and retrieval of memory (*Zars et al., 2000*; *Krashes et al., 2007*). Morphologically, both α/β and α'/β' neurons have axons with two branches that project dorsally and medially into respective α and β, or α' and β' lobes, whereas γ neurons send out single axon branches that project medially into the γ lobe (*Lee et al., 1999*). These three types of KCs can also be distinguished by differentially expressed marker genes. For instance, broad-complex, tramtrack, and bric-à-brac zinc finger transcription factor (BTBzf TF) Abrupt (Ab) is specifically expressed in γ neurons (*Liu et al., 2019*; *Lai et al., 2022*; *Hu et al., 1995*), whereas cell adhesion molecule Fasciclin

II and Rho guanine nucleotide exchange factor Trio are expressed in different subsets of KC types (*Liu et al., 2019*). Therefore, specifying KC types with proper cell numbers, unique cell identities, and distinct functions is a key aspect of MB formation.

Previous studies have revealed that the other BTBzf TF, Chronologically inappropriate morphogenesis (Chinmo), exhibits a graded expression pattern in KCs and plays a crucial role in diversifying main KC types (*Zhu et al., 2006*). Chinmo is highly expressed in early-born γ neurons, but its expression level is low in later-born α′/β′ and absent in α/β neurons (*Zhu et al., 2006*). Notably, Chinmo controls the expression of the third BTBzf TF, Maternal gene required for meiosis (Mamo), functioning to specify the identities of main KC types at the larval stage via a fine-tuning process. In particular, a high expression level of Chinmo inhibits Mamo expression in γ neurons, whereas low Chinmo expression promotes Mamo production in α′/β′ neurons (*Liu et al., 2019*). As a gradual reduction of Chinmo expression occurs during development, Mamo, adding to its function in α′/β′ neurons, takes over Chinmo's role of regulating the differentiation of γ neurons at the pupal stage (*Liu et al., 2019*; *Lai et al., 2022*; *Zhu et al., 2006*). In the absence of *chinmo* and *mamo*, molecular and morphological characteristics of γ and α′/β′ neurons are shifted toward those of α/β neurons, constituting a cell identity transformation (*Liu et al., 2019*; *Lai et al., 2022*; *Zhu et al., 2006*). Therefore, the late-born α/β neural identity is hypothesized to be a default status upon the loss of the neural identity determinants for early-born KCs (*Liu et al., 2019*; *Zhu et al., 2006*). However, it is also possible that key specification regulators of α/β neurons exist but have not yet been identified.

In this study, we leverage RNA-seq databases on KCs to identify type-specific markers as readouts for the cell identity of γ, α′/β′, and α/β neurons (*Alyagor et al., 2018*; *Shih et al., 2019*). By doing so, we identified a phylogenetically conserved Pipsqueak domain-containing TF, Ecdysone-induced protein 93F (Eip93F/E93), with preferential expression in α/β neurons. Loss of function of *E93* not only downregulated α/β-specific gene expression, including a subunit of T-type like voltage-gated calcium channel Ca-α1T, but it also upregulated γ-specific Ab in late-born KCs, implying that an identity shift toward early-born KCs had occurred. Intriguingly, RNAi knockdown of *E93* or *Ca-α1T* in α/β neurons further compromised animal behaviors, including foraging-related and night-time activities. In contrast, E93 overexpression precociously turned on Ca-α1T expression in early-born KCs at the expense of abolishing expression of early-born KC markers, such as Ab and Mamo. Notably, E93 was upregulated in early-born KCs in the absence of *chinmo* and *mamo* but diminished in late-born KCs upon Ab overexpression. Taken together, our results suggest that a hierarchical genetic network among *chinmo*, *mamo*, *E93*, and *ab* with potential feedback loops controls the identity and function of main KC types during the construction of functional MBs.

## Results

### Identification of KC-type-specific markers

By leveraging information from published RNA-seq studies showing preferentially expressed genes in adult KC types (*Alyagor et al., 2018*; *Shih et al., 2019*), we sought to identify a collection of KC-type-specific marker lines (available at stock centers) with GFP transgenes at genes of interest (*Venken et al., 2009*; *Venken et al., 2011*; *Morin et al., 2001*). The results of this screen for KC marker GFP lines are depicted in *Figure 1—figure supplement 1* and *Supplementary file 1*; highlights of certain KC-type-specific marker lines are described below. First, Abrupt (Ab)-GFP, a BAC clone-engineered GFP line, can be used to replace an excellent Ab antibody (generated by Dr. Crews laboratory but no longer available) for labeling γ neurons (*Liu et al., 2019*; *Lai et al., 2022*; *Hu et al., 1995*; *Figure 1A, B*, *Figure 1—figure supplement 2*). In addition, Lachesin (Lac; an Ig superfamily protein; *Llimargas et al., 2004*)-FSVS, a GFP-trapping line, expresses GFP-fused Lac specifically in α′/β′ neurons starting from the early pupal stage (*Figure 1C, D*, *Figure 1—figure supplement 3A*). Notably, we also identified two other GFP-trapping lines, Ecdysone-induced protein 93F (E93)-GFSTF and Calcium channel protein α1 subunit T (Ca-α1T)-GFSTF, that express GFP-fused proteins enriched in α/β neurons. Enrichment in these neurons is evidenced by patterns of complementary expression to Trio, a marker of γ and α′/β′ neurons (*Liu et al., 2019*; *Figure 1F, H*). Consistent with the notion that generation of most of α/β neurons occurs at the pupal stage (*Lee et al., 1999*), E93-GFSTF and Ca-α1T-GFSTF were not detectable in KCs at the wandering larval (WL) stage (*Figure 1E, G*, *Figure 1—figure supplement*

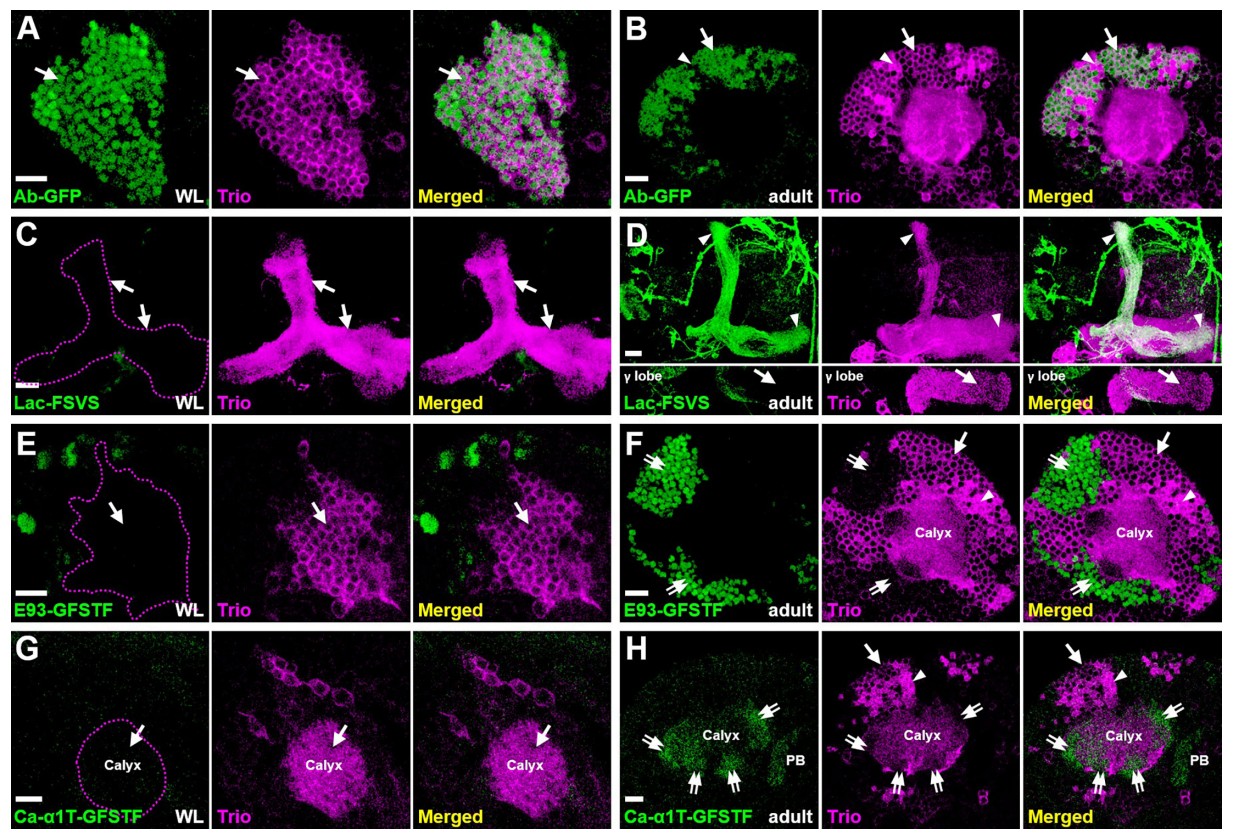

**Figure 1.** Expression patterns of main Kenyon cell (KC) type markers. (**A–H**) KC-type-specific GFP markers (green) were counter-stained with Trio (magenta) to reveal expression patterns at the wandering larval (WL) (**A**, **C**, **E**, **G**) and adult (**B**, **D**, **F**, **H**) stages. (**A, B**) Ab-GFP was primarily expressed in cell bodies of γ neurons at both WL and adult stages. The Trio signal indicates locations of γ neurons for staining observed only in cytosol (arrows) and α′/β′ neurons for staining in the entire cell (arrowheads). (**C, D**) Lac-FSVS expression was enriched in α′ and β′ lobes (arrowheads) of adult but not WL stage animals. The single section in the bottom panels of (**D**) reveals the lack of Lac-FSVS expression in the γ lobe. (**E–H**) E93-GFSTF and Ca-α1T-GFSTF were preferentially expressed in respective cell bodies and dendrites (the calyx) of α/β neurons (double-arrows) at adult but not WL stage animals. In addition to calyx expression, Ca-α1T-GFSTF was also seen in the protocerebral bridge (PB) of adult brains. Genotypes shown in all figures are summarized in *Supplementary file 2*. Scale bar: 10 μm.

The online version of this article includes the following figure supplement(s) for figure 1:

**Figure supplement 1.** GFP-line screen for Kenyon cell (KC) subtype markers.

**Figure supplement 2.** Downregulation of Ab-GFP in Kenyon cells (KCs) in the *chinmo* mutation.

**Figure supplement 3.** Early pupal expression of Lac-FSVS and E93-GFSTF in Kenyon cells (KCs).

*3B*). With this set of useful reagents, we set out to further explore the molecular mechanisms underlying cell identity specification of main KC types.

## Loss of function of *E93* suppresses the α/β-neural identity and has behavioral consequences

Since E93, a Pipsqueak-domain-containing TF, functions crucially in various biological processes (*Baehrecke and Thummel, 1995*; *Mundorf et al., 2019*; *Pahl et al., 2019*; *Lam et al., 2022*) and E93-GFSTF was preferentially expressed in α/β neurons (*Figure 1F*), we sought to investigate whether *E93* acts as a critical regulator for specifying the α/β-neural identity. First, we found that the expression of a α/β-specific marker, Ca-α1T-GFSTF, was diminished in the context of *E93* mutation (*E93^{Δ11}*) and in a line with *E93* knockdown due to overexpression of *E93* RNAi by a pan-KC driver, GAL4-OK107 (*Lee et al., 1999*; *Figure 2A, B*, *Figure 2—figure supplements 1 and 2*). In contrast, the same RNAi knockdown elicited no discernible effects on the expression levels of Trio and Lac-FSVS in γ and α′/β′ neurons (*Figure 2A, B*, *Figure 2—figure supplement 3*). In addition to the observed reduction

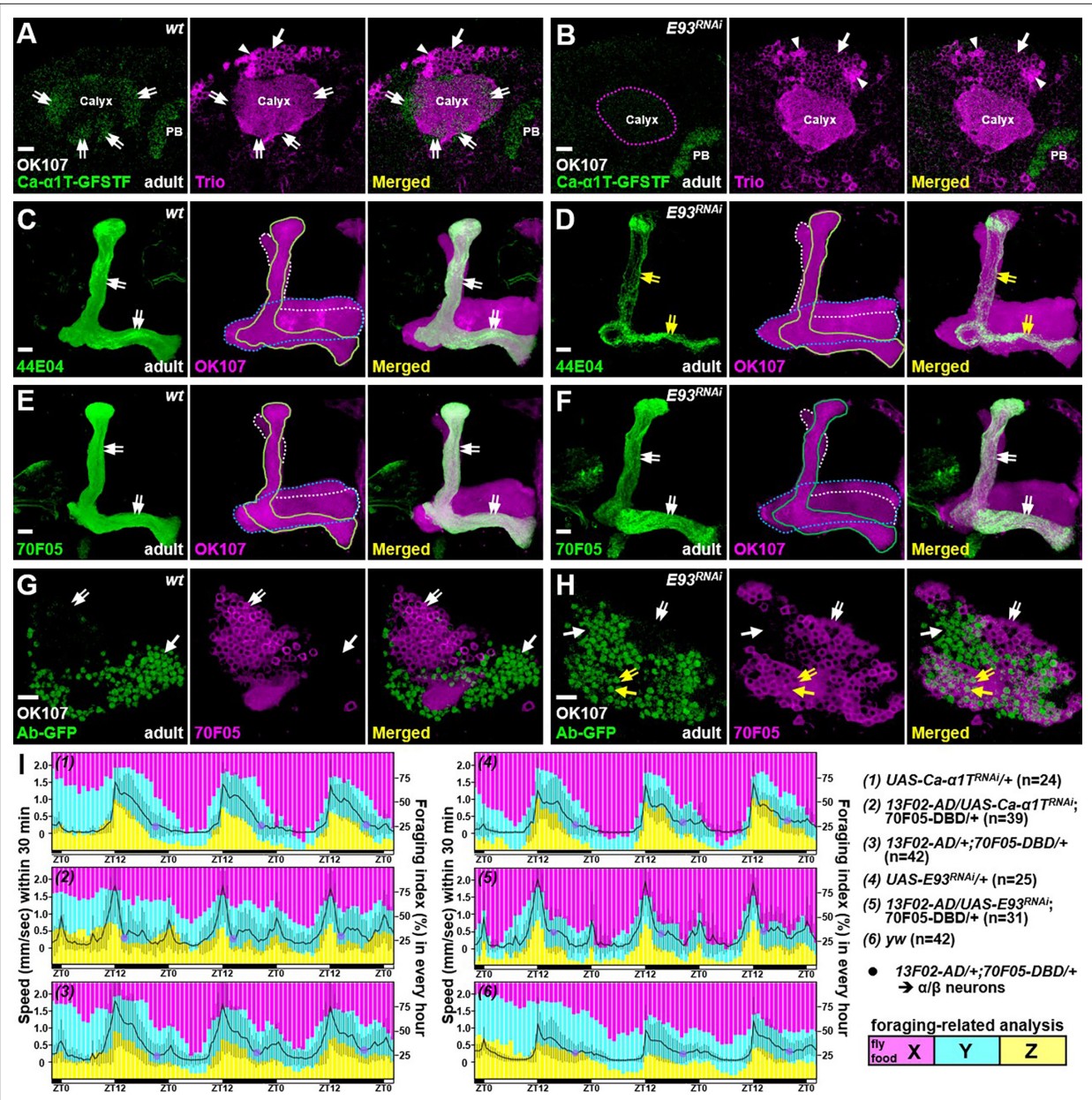

**Figure 2.** E93 specifies the α/β neural identity and affects animal behaviors. As compared to the wild-type controls (**A, C**), flies with overexpression of *E93* RNAi (**B, D**) by a pan-Kenyon cell (KC) driver, GAL4-OK107, had significantly impaired expression of Ca-α1T-GFSTF and 44E04-LexA in α/β neurons (double-arrows) in adult brains. As an internal control, the PB expression of Ca-α1T-GFSTF was intact under *E93* RNAi knockdown driven by GAL4-OK107. (**E, F**) *E93* knockdown did not block the expression of 70F05-LexA in α/β neurons to cause the detectable morphological defect in lobe regions (double-arrows). (**G, H**) However, compared to the wild-type, Ab-GFP expression was ectopically expressed in more than half of 70F05-LexA-positive neurons (double-arrows) when *E93* was knocked down in KCs. The expression levels of 44E04-LexA and 70F05-LexA were visualized by lexAop-myr-GFP in panels **C-F** and lexAop-mCD8::RFP in panels **G** and **H**. Lobes of γ, α/β, and α/β neurons were outlined in blue, white, and green, respectively, in panels **C-F**. Cell numbers of 70F05-LexA- and Ab-GFP-positive neurons were counted in *Figure 2—figure supplement 4*. Scale bar: 10 μm. (**I**) In control samples, including yw, *Ca-α1T* RNAi-, *E93* RNAi-, and α/β-neural driver (13F02-AD/70F05-DBD)-only flies, it took around 8 hr for the minimal speed (purple spots) to be reached at night. However, flies took around 2–4 hr to achieve a minimal speed when RNAi's for *Ca-α1T* and *E93* were overexpressed using 13F02-AD/70F05-DBD. Moving speed (black line) from the second day to the fifth day was calculated as the overall traveling distance (mm) for 30 min. Standard deviation (in gray) for each time point is shown. The bar graph depicts the duration of food region exploration (X to Z zones, from proximal to distal). All flies, except for *Ca-α1T* knockdown samples, tended to explore more in the X zone in the daytime. ZT: Zeitgeber time. The setting and analysis of the behavioral assay is detailed in *Figure 2—figure supplement 5*.

The online version of this article includes the following figure supplement(s) for figure 2:

*Figure 2 continued on next page*

*Figure 2 continued*

**Figure supplement 1.** Specificity of *E93* RNAi reagents in blocking the expression of E93-GFSTF.

**Figure supplement 2.** Downregulation of Ca-α1T-GFSTF in the *E93* mutation.

**Figure supplement 3.** *E93* RNAi knockdown does not affect the expression of Lac-FSVS.

**Figure supplement 4.** Statistical analysis of Ab-GFP expression in Kenyon cells (KCs) of wild-type and *E93* RNAi knockdown samples in *Figure 2H, I*.

**Figure supplement 5.** Setting, tracking, and related results for the behavioral assay.

of Ca-α1T-GFSTF, *E93* knockdown in KCs substantially compromised the expression of another α/β-specific marker, myr::GFP driven by 44E04-LexA (*Lai et al., 2022*; *Figure 2C, D*). Although these results seemed to indicate the loss of α/β neural identity due to loss of *E93* function, the findings also raised a possibility that the reduction of Ca-α1T-GFSTF and 44E04-LexA might simply be due to the absence of late-born KCs. However, this explanation was ruled out by the observation of relatively intact morphology of α and β lobes when we examined the expression of a third α/β-specific marker (myr::GFP driven by 70F05-LexA) upon *E93* knockdown in KCs (*Lai et al., 2022*; *Figure 2E, F*). More-over, compared to wild-type samples, the cell number of 70F05-LexA-positive neurons was not signifi-cantly altered in KCs when *E93* was knocked down (*Figure 2G, H*, *Figure 2—figure supplement 4*). Taking advantage of 70F05-LexA as a putative α/β-neural marker, we further found that more than half of the 70F05-LexA-positive neurons ectopically expressed γ-specific Ab-GFP in *E93* knockdown samples (*Figure 2G, H*, *Figure 2—figure supplement 4*), implying that α/β neurons had transformed into γ-like neurons. These results taken together suggested that E93, despite being unable to direct axon morphogenesis, is required for specifying KCs toward the cell identity of α/β neurons by regu-lating the expression of certain α/β-specific genes.

Since *Ca-α1T* encodes a subunit of a T-type like voltage-gated calcium channel, which regulates sleep behavior (*Jeong et al., 2015*), we wondered whether the loss of *E93* and *Ca-α1T* in α/β neurons could cause behavioral defects. To examine the behavior, we utilized a monitoring system to film and analyze the activities of individual flies (*Figure 2—figure supplement 5A, B*). The animal's speed of locomotion usually peaks around the day-to-night shift and gradually becomes reduced at night. In control samples (wild-type, RNAi-only, and GAL4-only lines), it took around 8 hr for the moving speed to reach a minimum during the night period (*Figure 2I*). Interestingly, night-time activity was significantly impaired (with a sharp reduction of moving speed to a minimal value at around 2–4 hr into the night period) when *Ca-α1T* and *E93* were knocked down using α/β neural drivers, 13F02-AD/70F05-DBD and c739-GAL4 (*Pavlowsky et al., 2024*; *Figure 2I*, *Figure 2—figure supplement 5C–E*). Although the overall moving speed was lower in samples of *E93* knockdown driven by c739-GAL4 than in samples of other genotypes, the wings of these *E93* knockdown flies appeared curly and aberrant, which might cause the movement defect. In addition to the effects on night-time activity, RNAi knock-down of *Ca-α1T* and *E93* seemed to compromise the foraging-related behavior (*Figure 2—figure supplement 5E*). Compared to control flies, *Ca-α1T*-deficient animals (especially for RNAi knockdown by c739-GAL4) tended to explore regions without fly food during the day-time period (*Figure 2I*, *Figure 2—figure supplement 5E*). Taken together, these results suggested that Ca-α1T (whose expression is regulated by E93) acts in α/β neurons to potentially control animal behavior.

### *E93* overexpression promotes α/β neural traits in early-born KCs

We next asked if E93 indeed plays a crucial role in specifying α/β neural identity, that is, does gain of E93 function transform the cell identity of other KC types to α/β neurons? In contrast to the undetect-able Ca-α1T-GFSTF expression in wild-type with two main KC types, γ and α'/β' neurons, at the WL stage, Ca-α1T-GFSTF expression was precociously upregulated in KCs when E93 was overexpressed by GAL4-OK107 (*Figure 3A, B*). Similarly, a portion of putative γ neurons exhibited α/β-neural like features (myr::GFP driven by R70F05-LexA) with axonal defects when E93 was overexpressed by the γ-neural driver, GAL4-201Y (*Lee et al., 1999*; *Figure 3C, D*). Consistent with this observation, we further found that E93 overexpression driven by GAL4-OK107 (but not Worniu-GAL4, a pan-neuroblast driver; *Pahl et al., 2019*) compromised the expression of specific markers in γ neurons at various developmental stages, including Ab-GFP, the receptor for insect molting hormone ecdysone (EcR-B1) and Mamo isoforms (*Figure 3E–J*, *Figure 3—figure supplements 1 and 2*). Along with the effect of impairing γ-specific marker expression, we further found that E93 overexpression driven

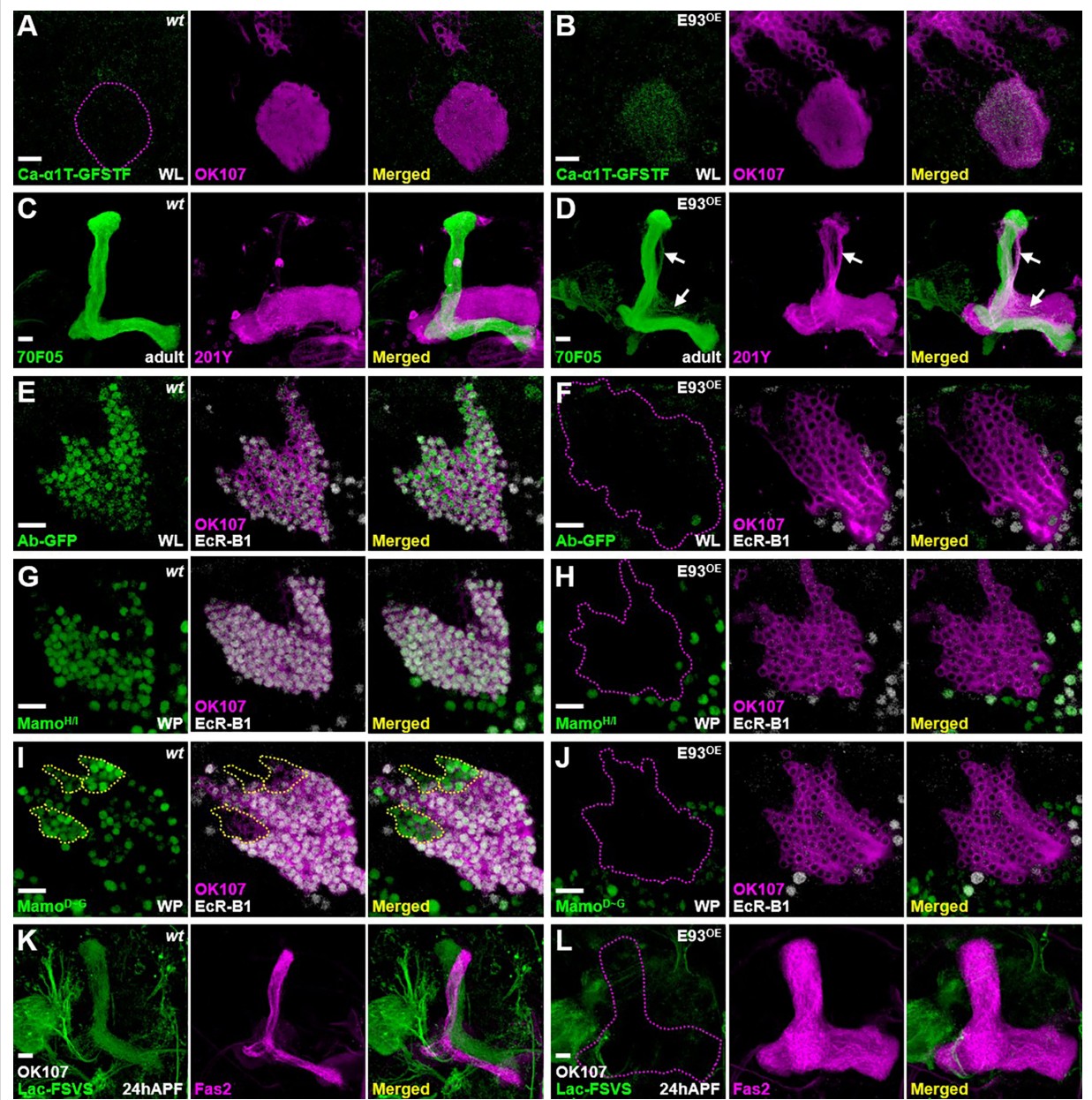

**Figure 3.** E93 is sufficient to shift the Kenyon cell (KC) identity toward α/β neural-like fate. (**A, B**) Overexpression of E93 driven by GAL4-OK107 caused precocious expression of α/β-specific Ca-α1T-GFSTF in early-born KCs at the wandering larval (WL) stage. (**C, D**) In addition, overexpression of E93 driven by a γ-neural driver, GAL4-201Y (magenta), ectopically turned on the expression of a α/β-specific 70F05-LexA driver in a portion of γ neurons (visualized by myr-GFP in green; arrow). On the other hand, overexpression of E93 abolished γ-specific markers, including Ab-GFP (**E, F**), Mamo^H/I (**G, H**), Mamo^D~G (weak green signal; **I, J**) and EcR-B1 (**E–J**), and α′/β′-specific Mamo^D~G (strong green signal within yellow dashed-line; **I, J**) in the early-born KCs at the white pupal (WP) stage. (**K, L**) E93 overexpression also compromised the Lac-FSVS expression in α′/β′ neurons and the morphology of mushroom body (MB) lobes revealed by cell adhesion molecule Fasciclin II (Fas2, strong magenta for labeling α and β lobes) at 24 hr after puparium formation (APF). An enhance-promoter (EP) line inserted at the proximal region of the *E93-A* 5′UTR was used to overexpress E93 in the gain-of-function experiments. The potency of the E93(EP) line was similar to two other in-house transgenic lines expressing E93-A and E93-B isoforms (see *Figure 3—figure supplement 1*). Scale bar: 10 μm.

The online version of this article includes the following figure supplement(s) for figure 3:

**Figure supplement 1.** *E93* gene and downregulation of Ab-GFP and EcR-B1 in Kenyon cells (KCs) by overexpression of E93 isoforms.

**Figure supplement 2.** E93 overexpression using a neuroblast driver does not cause defects in Kenyon cells (KCs).

by GAL4-OK107 impaired the expression of markers in α′/β′ neurons, including Mamo[D–G] isoforms and Lac-FSVS (*Figure 3I–L*). Its overexpression also caused morphological defects by perturbing the remodeling process that normally occurs in early-born KCs at 24 hr after puparium formation (APF; *Figure 3K, L*). These results taken together suggested that E93, when overexpressed, is sufficient to cause specification of KCs toward the α/β neural identity.

## Hierarchical genetic network of *chinmo*, *mamo*, *E93,* and *ab* controls the cell identity of main KC types

Since Chinmo and E93 act as key TFs to regulate the cell identity of main KC types, we wondered whether *chinmo* and *E93* might form a genetic network to diversify these KC types. Since *chinmo* controls the expression of γ-specific Ab (*Liu et al., 2019*; *Lai et al., 2022*; *Figure 1—figure supplement 2*), we decided to test whether the loss of Ab-GFP by E93 overexpression (seen in *Figure 3F*) was caused by compromised Chinmo expression. However, we did not observe a reduction of Chinmo level upon E93 overexpression at the first instar larval stage (the most abundant Chinmo expression stage in the wild-type; *Zhu et al., 2006*), even though the same manipulation did block Ab-GFP expression (*Figure 4A, B*). In contrast, E93-GFSTF expression was precociously upregulated in early-born KCs in a *chinmo* mutation line (*chinmo[1]*) and in the *chinmo* knockdown line (overexpression of *chinmo* RNAi driven by GAL4-OK107) at the WL stage (*Figure 4C, D*, *Figure 4—figure supplement 1*). In line with the facts that microRNA *let-7* and RNA-binding protein Syncrip (Syp) inhibit Chinmo expression (*Liu et al., 2015*; *Wu et al., 2012*), we further found that E93-GFSTF expression was abolished by *syp* RNAi knockdown and partially promoted by *let-7* overexpression (*Figure 4—figure supplement 2*). These results suggested that the adoption of γ neural identity by early-born KCs is in part due to suppression of the α/β-neural regulator E93 via Chinmo.

After the diminishment of Chinmo at the early pupal stage (*Zhu et al., 2006*), we wonder whether E93 would be disinhibited in γ neurons and whether their neural identity would be transformed if this disinhibition indeed occurs? Since γ neurons exist in adult brains and their neural identity is crucially regulated by Mamo at the pupal stage (*Lai et al., 2022*), we then tested whether Mamo takes over Chinmo's role to suppress E93 expression in γ neurons. As such, *mamo* RNAi knockdown indeed caused the upregulation of E93-GFSTF in KCs (within enriched Trio expression) of adult brains but not brains at the WL stage (*Figure 4E–G*), suggesting that Mamo could inhibit E93 expression in γ neurons to ensure their neural identity. Since the E93-mediated α/β neural identity is accompanied by absence of certain γ-specific traits, such as Ab-GFP (*Figures 2H and 3F*), we next sought to explore whether Ab also plays a crucial role in controlling the KC identity. Intriguingly, we found that the expression of α/β-specific Ca-α1T-GFSTF and E93-GFSTF was not detectable upon Ab overexpression (*Figure 4H, J*, *Figure 4—figure supplement 3*). This was accompanied by the expansion of Trio in the cytosol in almost all KCs, indicating the transformation of KCs into γ neuron-like cells. We also noted that RNAi knockdown of *ab* neither abolished Trio expression nor upregulated the expression of Ca-α1T-GFSTF caused by *E93* knockdown in early-born KCs (*Figure 4—figure supplements 4 and 5*). Nonetheless, these results together suggested that *chinmo*, *mamo*, *E93*, and *ab* form a hierarchical genetic network with potential feedback loops to control the identity of KCs (*Figure 5*).

## Discussion

Spatially and temporally expressed regulators are employed in the developing nervous system to generate a wide diversity of neuronal types. Spatial patterning cues, such as homeodomain-containing Hox proteins and Sonic Hedgehog, are well documented in the specification of motor neurons and interneurons in the spinal cord (*Jessell, 2000*). Beyond spatial identity, temporal regulation provides another axis of neuronal diversification: distinct neuron types are sequentially generated to form the six-layer laminar organization of the mammalian cerebral cortex, illustrating how layer-specific neurons arise from neural progenitors according to their birth order (*Kohwi and Doe, 2013*; *Kandel et al., 2000*). Similarly, in the *Drosophila* central nervous system, neural progenitors (neuroblasts) produce different neuron types in an invariant sequence under the control of temporal regulators (*Kohwi and Doe, 2013*; *Kandel et al., 2000*). Distinct sets of transcription factors expressed sequentially in neuroblasts of the embryonic ventral nerve cord and developing medulla direct fate specification, thereby generating diverse neurons for larval and visual circuit assembly (*Kohwi and Doe, 2013*; *Özel et al.,*

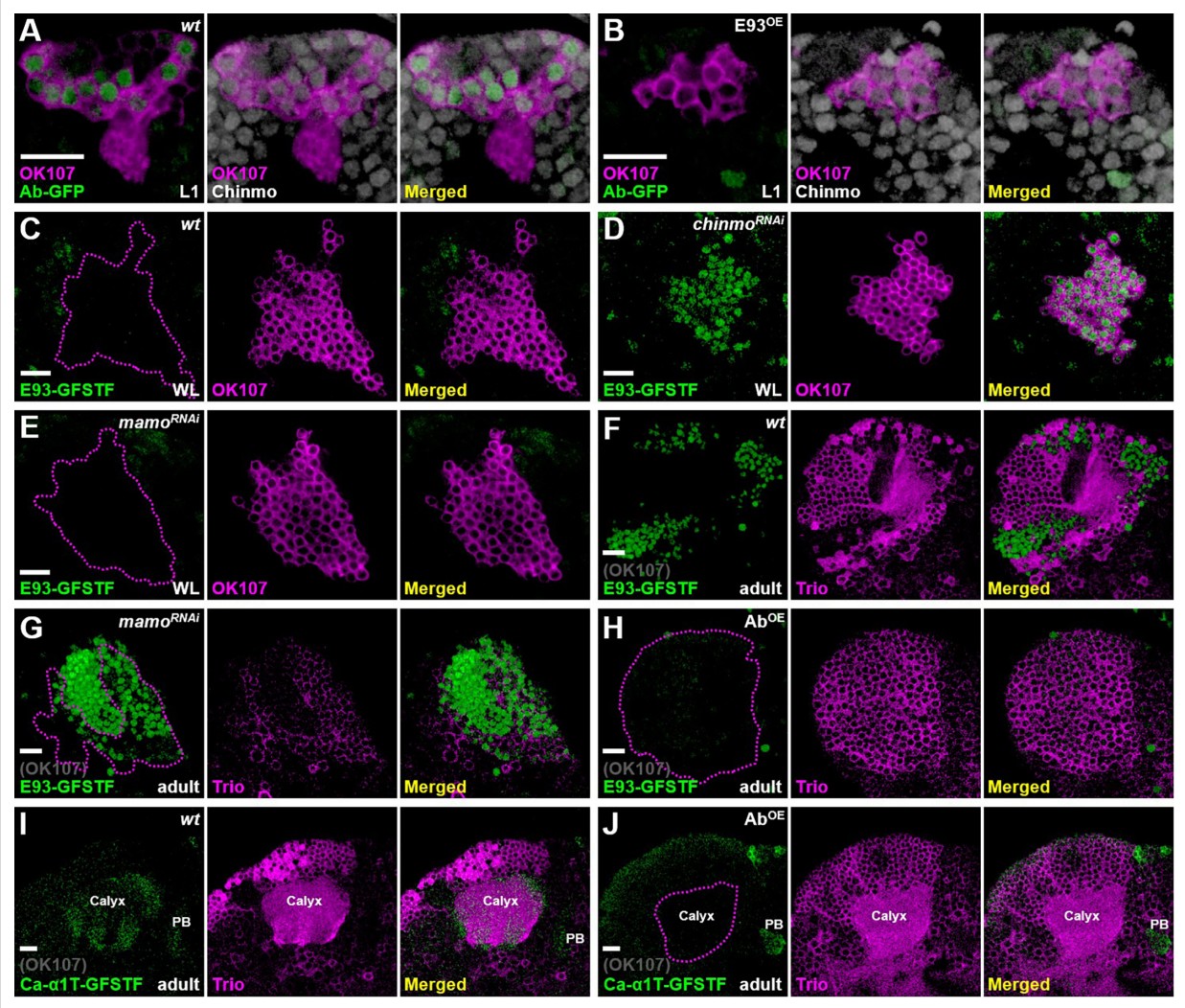

**Figure 4.** Genetic networks of *chinmo*, *mamo*, *E93*, and *ab* control Kenyon cell (KC) identity. (**A, B**) As compared to the wild-type, expression of Ab-GFP (green) was diminished in KCs at the first instar larval (L1) stage upon E93 overexpression driven by GAL4-OK107 (magenta). However, Chinmo expression (white) was not affected by E93 overexpression. (**C–E**) In contrast, RNAi knockdown of *chinmo*, but not *mamo*, driven by GAL4-OK107 (magenta) precociously turned on the expression of E93-GFSTF (green) in the early-born KCs at the wandering larval (WL) stage. (**F, G**) However, RNAi knockdown of *mamo* driven by GAL4-OK107 ectopically turned on expression of E93-GFSTF (green) in KCs with weak cytosolically expressed Trio (magenta) in adult brains (magenta dash lines). The weak Trio signal was possibly due to *mamo* RNAi knockdown in early-born KCs. E93-GFSTF was densely expressed in putative α/β neurons with negative Trio signal (region outside magenta dashed lines). (**H–J**) Ab overexpression driven by GAL4-OK107 diminished the expression of E93-GFSTF and Ca-α1T-GFSTF in KCs of adult brains. The Trio seemed to be expressed in the cytosol in almost all KCs upon Ab overexpression. Scale bar: 10 μm.

The online version of this article includes the following figure supplement(s) for figure 4:

**Figure supplement 1.** Precocious upregulation of E93-GFSTF in Kenyon cells (KCs) in the *chinmo* mutation.

**Figure supplement 2.** Overexpression of *let-7* and *syp* RNAi compromises the expression of E93-GFSTF in Kenyon cells (KCs).

**Figure supplement 3.** Overexpression of Ab compromises E93-GFSTP expression in Kenyon cells (KCs).

**Figure supplement 4.** RNAi knockdown of *ab* elicits no obvious effects on the expression of Trio, E93-GFSTF, and Ca-α1T-GFSTF in Kenyon cells (KCs).

**Figure supplement 5.** RNAi knockdown of *ab* fails to restore the Ca-α1T-GFSTF expression caused by E93 knockdown.

2021). In addition to sequential neuroblast regulation, temporal expression of transcription factors in postmitotic neurons also contributes to neuronal diversity by conferring distinct identities (*Zhu et al., 2006*; *Maurange et al., 2008*). For example, among the three sequentially generated KC types, γ, α′/β′, and α/β neurons, BTBzf TFs, Chinmo and Mamo, have been shown to play crucial roles in specifying early-born KCs as γ and α′/β′ neurons (*Lee et al., 1999*; *Liu et al., 2019*; *Zhu et al., 2006*). In this

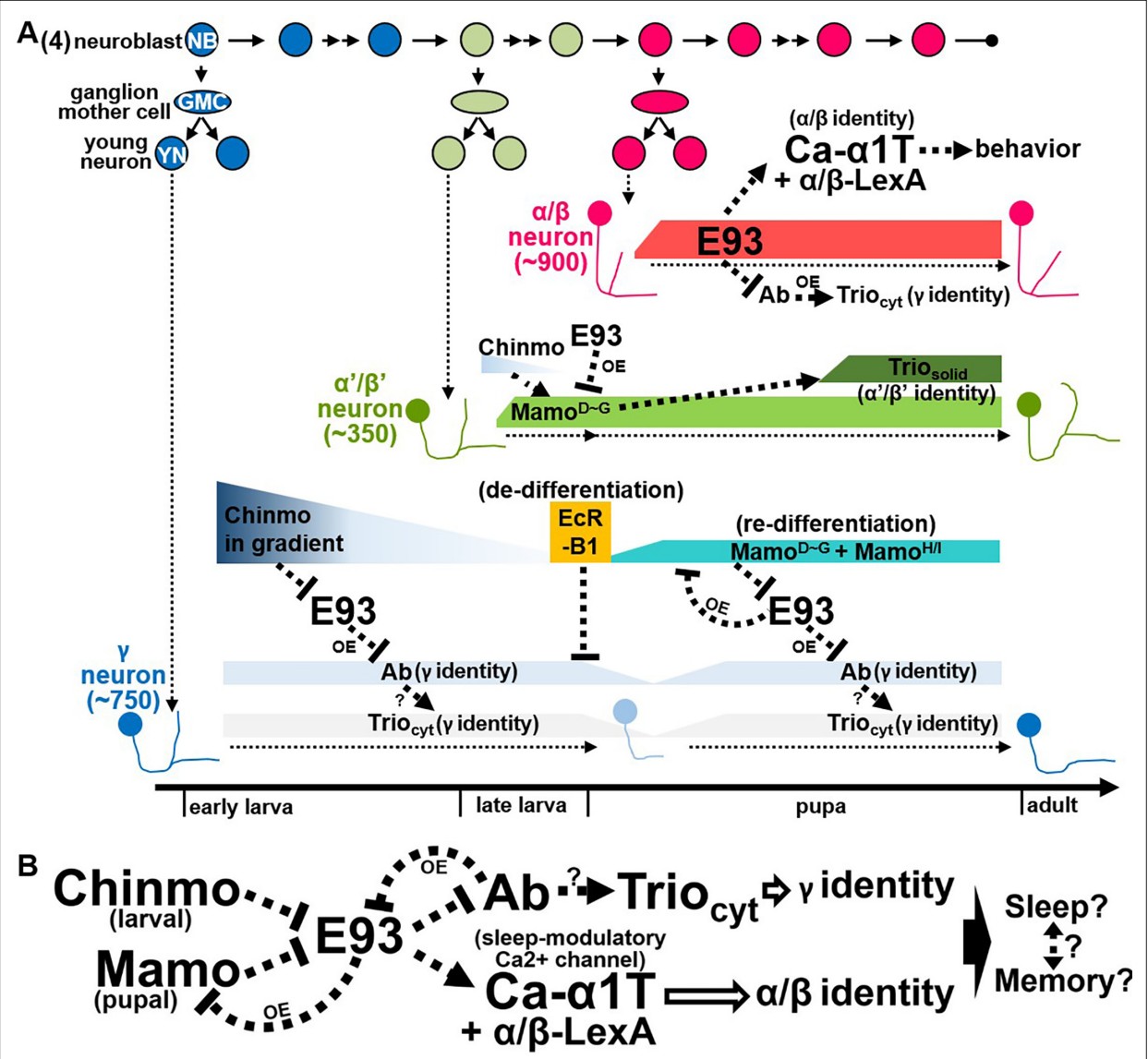

**Figure 5.** Hierarchical genetic networks govern the identity and function of Kenyon cells (KCs) in the construction of Mbs. (**A, B**) Scheme delineates hierarchical genetic networks among *chinmo*, *mamo*, *E93*, and *ab* with feedback loops that control the cell identities of γ and α′/β′ neurons. Syp and *let-7* are included in genetic networks to potentially link the regulation of E93 and sleeping modulatory calcium channel Ca-α1T (*Liu et al., 2015*; *Wu et al., 2012*; *Goodwin et al., 2018*; *Figure 4—figure supplement 2*). Based on the results of gain-of-function studies (*Figures 3H, J and 4H*), possible feedback regulation in the genetic network is indicated with OE (as the abbreviation of overexpression). Since E93 regulates the Ca-α1T expression (*Figure 2B*) and since *let-7* is also crucial for the sleep behavior (*Goodwin et al., 2018*), E93 and Ca-α1T may be potentially associated with sleep and memory behaviors through KCs. Question marks (?) indicate possible regulation in the genetic network.

study, we revealed the molecular mechanisms of KC specification by identifying and characterizing Pipsqueak domain-containing TF E93 as a driver of KC identity toward α/β neurons. Our findings that E93 regulates the expression of calcium channel Ca-α1T to subsequently control animal behaviors also provide an illustration that neural identity specification is associated with acquisition of specific functions among main KC types. We further showed that the specification of KC types is controlled via a genetic network formed by *chinmo*, *mamo*, *E93*, and *ab*, which mainly controls the KC identity of γ and α/β neurons during construction of the adult MB (*Figure 5*).

Despite that reciprocal regulation between BTBzf TFs and E93 has been reported in multiple cell types during the *Drosophila* development (*Truman and Riddiford, 2022*; *Cruz et al., 2024*), here, we disclosed a novel strategy of using genetic networks of *BTBzf TFs* and *E93* to specify distinct cell identities among neurons derived from the same neuroblasts (*Figure 5*). To diversify KC types,

Chinmo and Mamo take turns to inhibit the E93 expression, thereby establishing the cell identity of γ neurons, as suggested by Ab expression (*Liu et al., 2019*; *Figures 2H and 4G*). Without the inhibitory effects of Chinmo and Mamo on E93 expression, late-born KCs specify into α/β neurons by turning on the expression of α/β-specific genes and suppressing the expression of γ-specific genes, Ab included (*Figure 2*). Since E93 is an ecdysone-induced protein and Mamo expression is primed by ecdysone signaling in γ neurons (*Lai et al., 2022*; *Baehrecke and Thummel, 1995*), blocking EcR-B1 is expected to enhance E93 expression in these neurons. By the fact that overexpression of E93 inhibited EcR-B1 expression in γ neurons (*Figure 3F, H, J*), future investigations should be able to unravel the intricate regulatory interplay between E93 and ecdysone signaling. In addition, since *E93* knockdown did not alter the lobe morphology of α/β neurons (*Figure 2F*), it is also possible that unidentified regulators might work in concert with E93 to specify the cell fate of α/β neurons. Although how the genetic network of *BTBzf TFs* and *E93* specifies α′/β′ neural identity has not yet been fully resolved, the Mamo expression promoted by the low level of Chinmo, which is regulated by the transforming growth factor beta signaling in MB neuroblasts, is crucial for KCs to adopt the α′/β′ neural identity (*Liu et al., 2019*; *Rossi and Desplan, 2020*). Intriguingly, our results revealed that E93 overexpression can abolish the expression of Mamo to block KCs from adopting the α′/β′ neural identity (*Figure 3J, L*). All these results taken together portray that the genetic network of *BTBzf TFs* and *E93* is very likely to control the KC identity during construction of the adult MB. In light of the phylogenetic conservation of the *E93* family and *tramtrack* (*TTK*)-type *BTBzf TFs*, including *chinmo*, *mamo*, and *ab*, as Arthropoda-specific genes (*Bonchuk et al., 2024*; *Takayanagi-Kiya et al., 2017*), one cannot help but wonder whether *TTK*-type *BTBzf TF*s might have been evolutionarily introduced to intervene with *E93* to regulate the development of multiple arthropodan tissues; perhaps future investigations may delineate their relationship on neuron-type diversification in the nervous system.

Once the identities of neuron types are determined by key regulators, how do the cells acquire specific functions in the nervous system to eventually participate in animal behaviors? Previous findings, together with our current studies, might shed some light on this topic. First, the expression of Chinmo and Ab is inhibited in late-born KCs, in part due to the preferential expression of *let-7* in α/β neurons (*Wu et al., 2012*; *Kucherenko et al., 2012*). In addition, the neuronal (and KC) contribution of *let-7* regulates day- and night-time sleeping behaviors in developmental- and adult-restricted manners (*Goodwin et al., 2018*). Furthermore, Chinmo and *let-7* potentially regulate the expression of E93, for which expression in α/β neurons governs the expression of a phylogenetically conserved sleep-modulatory calcium channel Ca-α1T to regulate animal behaviors (*Jeong et al., 2015*; *Figures 2B and 4D*, *Figure 4—figure supplement 2A*). Interestingly, blocking neurotransmission with tetanus toxin light chain (TeNT-Ln) in the MB has been shown to promote locomotion, an outcome opposite to the *Ca-α1T* knockdown results observed in our study (*Martin et al., 1998*; *Figure 2I*, *Figure 2—figure supplement 5E*). This discrepancy may stem from TeNT-Ln acting on MB axons to inhibit downstream neurons, whereas the effect of *Ca-α1T* knockdown primarily occurs in MB dendrites that receive inputs from upstream neurons (*Martin et al., 1998*; *Figure 2I*). These results, together with the previously reported function of E93 in circadian rhythm, provide an intriguing link between E93 and sleep-associated behaviors (*Yip et al., 2024*). Since KC types were reported to function crucially in sleep/arousal and alcohol-induced sleep deficit behaviors (*Sengupta et al., 2019*; *Draper et al., 2024*; *Chvilicek et al., 2025*), our finding of KC identity specification and Ca-α1T expression under the regulation of E93 suggests that this process may in some way endow α/β neurons with the capacity to control animal behaviors related to sleep. Since the intricate relationship between sleep and memory is well established (*Walker et al., 2002*; *Graves et al., 2003*; *Seugnet et al., 2008*; *Li et al., 2009*), future deeper investigations on cell-type specification of memory-associated neurons are expected to provide insights into how to acquire unique functions among neurons to regulate these two crucial traits of animals.

# Materials and methods

## Key resources table

| Reagent type (species) or resource | Designation | Source or reference | Identifiers | Additional information |
|---|---|---|---|---|
| Genetic reagent (*D. melanogaster*) | Ab-GFP | Bloomington *Drosophila* Stock Center | BDSC:38626; FLYB: FBti0147714; RRID:BDSC_38626 | FlyBase symbol: PBac{ab-GFP.FLAG}VK00033 |
| Genetic reagent (*D. melanogaster*) | Lac-FSVS | Kyoto *Drosophila* Resource Center | DGGR:115308; FLYB: FBti0143795 | FlyBase symbol: PBac{769.FSVS-1}Lac$^{CPTI002601}$ |
| Genetic reagent (*D. melanogaster*) | E93-GFSTF | Bloomington *Drosophila* Stock Center | BDSC:59412; FLYB: FBti0178367; RRID:BDSC_59412 | FlyBase symbol: Mi{PT-GFSTF.1}Eip93F$^{MI05200-GFSTF.1}$ |
| Genetic reagent (*D. melanogaster*) | Ca-α1T-GFSTF | Bloomington *Drosophila* Stock Center | BDSC:61800; FLYB: FBti0178427; RRID:BDSC_61800 | FlyBase symbol: Mi{PT-GFSTF.0}Ca-α1T$^{MI08565-GFSTF.0}$ |
| Genetic reagent (*D. melanogaster*) | hs-FLP[12], UAS-mCD8::GFP | Bloomington *Drosophila* Stock Center | BDSC:28832; RRID:BDSC_28832 | FlyBase symbol: n.a. |
| Genetic reagent (*D. melanogaster*) | tubP-GAL80,FRT$^{40A}$ | Bloomington *Drosophila* Stock Center | BDSC:5192; RRID:BDSC_5192 | FlyBase symbol: n.a. |
| Genetic reagent (*D. melanogaster*) | chinmo$^{1}$,FRT$^{40A}$ | Bloomington *Drosophila* Stock Center | BDSC:59969; RRID:BDSC_59969 | FlyBase symbol: n.a |
| Genetic reagent (*D. melanogaster*) | GAL4-OK107 | Bloomington *Drosophila* Stock Center | BDSC:854; FLYB: FBal0242600; RRID:BDSC_854 | FlyBase symbol: ey$^{OK107}$ |
| Genetic reagent (*D. melanogaster*) | UAS-mCD8::RFP | Bloomington *Drosophila* Stock Center | BDSC:32219; FLYB: FBti0131967; RRID:BDSC_32219 | FlyBase symbol: P{10XUAS-IVS-mCD8::RFP}attP40 |
| Genetic reagent (*D. melanogaster*) | UAS-mCD8::RFP | Bloomington *Drosophila* Stock Center | BDSC:32218; FLYB: FBti0131950; RRID:BDSC_32218 | FlyBase symbol: P{10XUAS-IVS-mCD8::RFP}attP2 |
| Genetic reagent (*D. melanogaster*) | UAS-E93-RNAi$^{BDSC57868}$ | Bloomington *Drosophila* Stock Center | BDSC:57868; FLYB: FBti0164035; RRID:BDSC_57868 | FlyBase symbol: P{TRiP.HMC04773}attP40 |
| Genetic reagent (*D. melanogaster*) | UAS-E93-RNAi$^{VDRC104390}$ | Vienna *Drosophila* Resource Center | VDRC:104390; FLYB: FBti0120934 | FlyBase symbol: P{KK108140}VIE-260B |
| Genetic reagent (*D. melanogaster*) | UAS-Ca-α1T-RNAi | Bloomington *Drosophila* Stock Center | BDSC:39029; FLYB: FBti0149691; RRID:BDSC_39029 | FlyBase symbol: P{TRiP.HMS01948}attP40 |
| Genetic reagent (*D. melanogaster*) | GAL4-c739 | Bloomington *Drosophila* Stock Center | BDSC:7362; FLYB: FBti0002926; RRID:BDSC_7362 | FlyBase symbol: P{GawB}Hr39$^{c739}$ |
| Genetic reagent (*D. melanogaster*) | 44E04-LexA::P65 | Bloomington *Drosophila* Stock Center | BDSC:52736; FLYB: FBti0155872; RRID:BDSC_52736 | FlyBase symbol: P{GMR44E04-lexA}attP40 |
| Genetic reagent (*D. melanogaster*) | 70F05-LexA::P65 | Bloomington *Drosophila* Stock Center | BDSC:53629; FLYB: FBti0156295; RRID:BDSC_53629 | FlyBase symbol: P{GMR70F05-lexA}attP40 |
| Genetic reagent (*D. melanogaster*) | 13F02-p65.AD | Bloomington *Drosophila* Stock Center | BDSC:89699; FLYB: FBti0187130; RRID:BDSC_89699 | FlyBase symbol: P{R13F02-p65.AD}attP40 |
| Genetic reagent (*D. melanogaster*) | 70F05-GAL4.DBD | Bloomington *Drosophila* Stock Center | BDSC:69380; FLYB: FBti0191783; RRID:BDSC_69380 | FlyBase symbol: P{R70F05-GAL4.DBD}attP2 |
| Genetic reagent (*D. melanogaster*) | LexAop2-myr::GFP [VK5] | Rubin lab; *Venken et al., 2009* | n.a. | FlyBase symbol: n.a. |
| Genetic reagent (*D. melanogaster*) | LexAop2-mCD8::RFP [attP2] | Rubin lab; *Venken et al., 2009* | n.a. | FlyBase symbol: n.a. |
| Genetic reagent (*D. melanogaster*) | FRT$^{82B}$,tubP-GAL80 | Bloomington *Drosophila* Stock Center | BDSC:5135; RRID:BDSC_5135 | FlyBase symbol: n.a. |
| Genetic reagent (*D. melanogaster*) | FRT$^{82B}$ | Bloomington *Drosophila* Stock Center | BDSC:86313; FLYB: FBti0002074; RRID:BDSC_86313 | FlyBase symbol: P{neoFRT}82B |
| Genetic reagent (*D. melanogaster*) | E93$^{Δ11}$ | Bloomington *Drosophila* Stock Center | BDSC:93128; FLYB: FBal0369310; RRID:BDSC_93128 | FlyBase symbol: Eip93F$^{Δ11}$ |
| Genetic reagent (*D. melanogaster*) | E93(EP) | Bloomington *Drosophila* Stock Center | BDSC:30179; FLYB: FBti0128429; RRID:BDSC_30179 | FlyBase symbol: P{EP}Eip93F$^{G7133}$ |
| Genetic reagent (*D. melanogaster*) | UAS- E93-A [VK37] | Yu lab; this study | n.a. | FlyBase symbol: n.a. |

*Continued on next page*

*Continued*

| Reagent type (species) or resource | Designation | Source or reference | Identifiers | Additional information |
|---|---|---|---|---|
| Genetic reagent (*D. melanogaster*) | UAS- E93-B [VK37] | Yu lab; this study | n.a. | FlyBase symbol: n.a. |
| Genetic reagent (*D. melanogaster*) | GAL4-201Y | Bloomington *Drosophila* Stock Center | BDSC:4440; FLYB: FBti0002924; RRID:BDSC_4440 | FlyBase symbol: P{GawB}Tab2[201Y] |
| Genetic reagent (*D. melanogaster*) | mamo[H/I]-HA | Yu lab; **Venken et al., 2011** | n.a. | FlyBase symbol: n.a. |
| Genetic reagent (*D. melanogaster*) | Mamo[D~G]-HA | Yu lab; **Venken et al., 2011** | n.a. | FlyBase symbol: n.a. |
| Genetic reagent (*D. melanogaster*) | UAS-chinmo-RNAi [VK37] | Yu lab; **Liu et al., 2019** | n.a. | FlyBase symbol: n.a. |
| Genetic reagent (*D. melanogaster*) | UAS-LUC-let7 [attp2] | Bloomington *Drosophila* Stock Center | BDSC:41171; FLYB: FBti0148655; RRID:BDSC_41171 | FlyBase symbol: P{UAS-LUC-mir-let7.T}attP2 |
| Genetic reagent (*D. melanogaster*) | UAS-syp-RNAi | Vienna *Drosophila* Resource Center | VDRC:33011; FLYB: FBti0098886 | FlyBase symbol: P{GD9477}v33011. |
| Genetic reagent (*D. melanogaster*) | UAS-mamo-RNAi | Bloomington *Drosophila* Stock Center | BDSC:44103; FLYB: FBti0158705; RRID:BDSC_44103 | FlyBase symbol: P{TRiP.HMS02823}attP40 |
| Genetic reagent (*D. melanogaster*) | UAS-ab | Bloomington *Drosophila* Stock Center | BDSC:23639; FLYB: FBti0077844; RRID:BDSC_23639 | FlyBase symbol: P{UAS-ab.B}55 |
| Genetic reagent (*D. melanogaster*) | UAS-ab-HA [ZH-86Fb] | Zurich ORFeome Project | FlyORF:000705; FLYB: FBti0161305 | FlyBase symbol: M{UAS-ab.ORF.3xHA.GW}ZH-86Fb |
| Genetic reagent (*D. melanogaster*) | UAS-Dcr2 | Bloomington *Drosophila* Stock Center | BDSC:24651; FLYB: FBti0100276; RRID:BDSC_24651 | FlyBase symbol: P{UAab S-Dcr-2.D}10 |
| Genetic reagent (*D. melanogaster*) | UAS-ab-RNAi | Vienna *Drosophila* Resource Center | VDRC:104582; FLYB: FBti0122240 | FlyBase symbol: P{KK110195}VIE-260B |
| Genetic reagent (*D. melanogaster*) | dan-GFP | Bloomington *Drosophila* Stock Center | BDSC:92324; FLYB: FBti0214343; RRID:BDSC_92324 | FlyBase symbol: P{dan-GFP.FPTB}attP40 |
| Genetic reagent (*D. melanogaster*) | dlp-GFSTF | Bloomington *Drosophila* Stock Center | BDSC:60540; FLYB: FBti0178451; RRID:BDSC_60540 | FlyBase symbol: Mi{PT-GFSTF.1}dlp[MI04217-GFSTF.1] |
| Genetic reagent (*D. melanogaster*) | ed-GFSTF | Bloomington *Drosophila* Stock Center | BDSC:59777; FLYB: FBti0178373; RRID:BDSC_59777 | FlyBase symbol: Mi{PT-GFSTF.1}ed[MI01552-GFSTF.1] |
| Genetic reagent (*D. melanogaster*) | Imp-SVS | Kyoto *Drosophila* Resource Center | DGGR:115455; FLYB: FBti0143574 | FlyBase symbol: PBac{802.P.SVS-2}Imp[CPTI003910] |
| Genetic reagent (*D. melanogaster*) | SIFaR-GFSTF | Bloomington *Drosophila* Stock Center | BDSC:60228; FLYB: FBti0178529; RRID:BDSC_60228 | FlyBase symbol: Mi{PT-GFSTF.1}SIFaR[MI05376-GFSTF.1] |
| Genetic reagent (*D. melanogaster*) | TkR86C-GFSTF | Bloomington *Drosophila* Stock Center | BDSC:60549; FLYB: FBti0178567; RRID:BDSC_60549 | FlyBase symbol: Mi{PT-GFSTF.2}TkR86C[MI05788-GFSTF.2] |
| Genetic reagent (*D. melanogaster*) | CG31637-GFSTF | Bloomington *Drosophila* Stock Center | BDSC:64438; FLYB: FBti0181868; RRID:BDSC_64438 | FlyBase symbol: Mi{PT-GFSTF.2}CG31637[MI03598-GFSTF.2] |
| Genetic reagent (*D. melanogaster*) | CG43373-GFSTF | Bloomington *Drosophila* Stock Center | BDSC:60239 FLYB: FBti0178336; RRID:BDSC_60239 | FlyBase symbol: Mi{PT-GFSTF.0}CG43373[MI05926-GFSTF.0] |
| Genetic reagent (*D. melanogaster*) | CG4404-GFP | Bloomington *Drosophila* Stock Center | BDSC:90835 FLYB: FBti0212634; RRID:BDSC_90835 | FlyBase symbol: P{CG4404-GFP.FPTB}attP40 |
| Genetic reagent (*D. melanogaster*) | crb-GFSTF | Bloomington *Drosophila* Stock Center | BDSC:61781 FLYB: FBti0178594; RRID:BDSC_61781 | FlyBase symbol: Mi{PT-GFSTF.0}crb[MI05382-GFSTF.0] |
| Genetic reagent (*D. melanogaster*) | Lmpt-GFSTF | Bloomington *Drosophila* Stock Center | BDSC:66776 FLYB: FBti0185324; RRID:BDSC_66776 | FlyBase symbol: Mi{PT-GFSTF.2}Lmpt[MI04319-GFSTF.2] |
| Genetic reagent (*D. melanogaster*) | mbc-SVS | Kyoto *Drosophila* Resource Center | DGGR:115505; FLYB: FBti0143988 | FlyBase symbol: PBac{602.P.SVS-1}mbc[CPTI001082] |
| Genetic reagent (*D. melanogaster*) | Octbeta3R-GFSTF | Bloomington *Drosophila* Stock Center | BDSC:60245; FLYB: FBti0178463; RRID:BDSC_60245 | FlyBase symbol: Mi{PT-GFSTF.1}Octβ3R[MI06217-GFSTF.1] |
| Genetic reagent (*D. melanogaster*) | Ace-GFSTF | Bloomington *Drosophila* Stock Center | BDSC:60260; FLYB: FBti0178684; RRID:BDSC_60260 | FlyBase symbol: Mi{PT-GFSTF.1}Ace[MI07345-GFSTF.1] |

*Continued on next page*

*Continued*

| Reagent type (species) or resource | Designation | Source or reference | Identifiers | Additional information |
|---|---|---|---|---|
| Genetic reagent (*D. melanogaster*) | app-GFSTF | Bloomington *Drosophila* Stock Center | BDSC:60283; FLYB: FBti0178413; RRID:BDSC_60283 | FlyBase symbol: Mi{PT-GFSTF.0}app[MI11129-GFSTF.0] |
| Genetic reagent (*D. melanogaster*) | beat-IV-GFSTF | Bloomington *Drosophila* Stock Center | BDSC:66506; FLYB: FBti0178471; RRID:BDSC_66506 | FlyBase symbol: Mi{PT-GFSTF.1}beat-IV[MI05715-GFSTF.1] |
| Genetic reagent (*D. melanogaster*) | Ccn-GFSTF | Bloomington *Drosophila* Stock Center | BDSC:60259; FLYB: FBti0178562; RRID:BDSC_60259 | FlyBase symbol: Mi{PT-GFSTF.1}Ccn[MI06971-GFSTF.1] |
| Genetic reagent (*D. melanogaster*) | CG4829-FSVS | Kyoto *Drosophila* Resource Center | DGGR:115623; FLYB: FBti0143506 | FlyBase symbol: PBac{810.P.FSVS-2}CG4829[CPTI004450] |
| Genetic reagent (*D. melanogaster*) | Cyp4p3-GFSTF | Bloomington *Drosophila* Stock Center | BDSC:59829; FLYB: FBti0187664; RRID:BDSC_59829 | FlyBase symbol: Mi{PT-GFSTF.1}hig[MI05774-GFSTF.1m] |
| Genetic reagent (*D. melanogaster*) | DAT-sfGFP | Vienna *Drosophila* Resource Center | VDRC:318840; FLYB: FBti0198419 | FlyBase symbol: PBac{fTRG01319.sfGFP-TVPTBF}VK00033 |
| Genetic reagent (*D. melanogaster*) | dnr1-GFSTF | Bloomington *Drosophila* Stock Center | BDSC:76236; FLYB: FBti0185341; RRID:BDSC_76236 | FlyBase symbol: Mi{PT-GFSTF.0}dnr1[MI01678-GFSTF.0] |
| Genetic reagent (*D. melanogaster*) | dpr17-GFSTF | Bloomington *Drosophila* Stock Center | BDSC:61801; FLYB: FBti0178315; RRID:BDSC_61801 | FlyBase symbol: Mi{PT-GFSTF.1}dpr17[MI08707-GFSTF.1] |
| Genetic reagent (*D. melanogaster*) | Epac-GFSTF | Bloomington *Drosophila* Stock Center | BDSC:66364; FLYB: FBti0183610; RRID:BDSC_66364 | FlyBase symbol: Mi{PT-GFSTF.0}Epac[MI06245-GFSTF.0] |
| Genetic reagent (*D. melanogaster*) | eys-GFSTF | Bloomington *Drosophila* Stock Center | BDSC:63162; FLYB: FBti0180153; RRID:BDSC_63162 | FlyBase symbol: Mi{PT-GFSTF.2}eys[MI01874-GFSTF.2] |
| Genetic reagent (*D. melanogaster*) | fz3-sfGFP | Vienna *Drosophila* Resource Center | VDRC:318166; FLYB: FBti0198654 | FlyBase symbol: PBac{fTRG00593.sfGFP-TVPTBF}VK00033 |
| Genetic reagent (*D. melanogaster*) | igl-GFSTF | Bloomington *Drosophila* Stock Center | BDSC:60527; FLYB: FBti0178491; RRID:BDSC_60527 | FlyBase symbol: Mi{PT-GFSTF.1}igl[MI02290-GFSTF.1] |
| Genetic reagent (*D. melanogaster*) | LRP1-GFSTF | Bloomington *Drosophila* Stock Center | BDSC:60248; FLYB: FBti0178454; RRID:BDSC_60248 | FlyBase symbol: Mi{PT-GFSTF.1}LRP1[MI06376-GFSTF.1] |
| Genetic reagent (*D. melanogaster*) | mamo-sfGFP | Vienna *Drosophila* Resource Center | VDRC:318601; FLYB: FBti0198943 | FlyBase symbol: PBac{fTRG00552.sfGFP-TVPTBF}VK00033 |
| Genetic reagent (*D. melanogaster*) | Mp-GFSTF | Bloomington *Drosophila* Stock Center | BDSC:60567; FLYB: FBti0178435; RRID:BDSC_60567 | FlyBase symbol: Mi{PT-GFSTF.0}Mp[MI09316-GFSTF.0] |
| Genetic reagent (*D. melanogaster*) | msi-GFSTF | Bloomington *Drosophila* Stock Center | BDSC:61750; FLYB: FBti0178348; RRID:BDSC_61750 | FlyBase symbol: Mi{PT-GFSTF.2}msi[MI00977-GFSTF.2] |
| Genetic reagent (*D. melanogaster*) | Ndae1-GFSTF | Bloomington *Drosophila* Stock Center | BDSC:61778; FLYB: FBti0178493; RRID:BDSC_61778 | FlyBase symbol: Mi{PT-GFSTF.2}Ndae1[MI05100-GFSTF.2] |
| Genetic reagent (*D. melanogaster*) | nuf-GFSTF | Bloomington *Drosophila* Stock Center | BDSC:61802; FLYB: FBti0178615; RRID:BDSC_61802 | FlyBase symbol: Mi{PT-GFSTF.2}nuf[MI09643-GFSTF.2] |
| Genetic reagent (*D. melanogaster*) | rhea-GFSTF | Bloomington *Drosophila* Stock Center | BDSC:39649; FLYB: FBti0147808; RRID:BDSC_39649 | FlyBase symbol: Mi{PT-GFSTF.0}rhea[MI00296-GFSTF.0] |
| Genetic reagent (*D. melanogaster*) | smal-sfGFP | Vienna *Drosophila* Resource Center | VDRC:318203; FLYB: FBti0198848 | FlyBase symbol: PBac{fTRG00715.sfGFP-TVPTBF}VK00033 |
| Genetic reagent (*D. melanogaster*) | tok-GFSTF | Bloomington *Drosophila* Stock Center | BDSC:60550; FLYB: FBti0178631; RRID:BDSC_60550 | FlyBase Mi{PT-GFSTF.1}tok[MI06118-GFSTF.1] |
| Genetic reagent (*D. melanogaster*) | Zasp67-sfGFP | Vienna *Drosophila* Resource Center | VDRC:318355; FLYB: FBti0198786 | FlyBase symbol: PBac{fTRG01384.sfGFP-TVPTBF}VK00033 |
| Antibody | anti-Fas2 (Mouse monoclonal) | Developmental Studies Hybridoma Bank | Cat# AB_528235, RRID:AB_528235 | IF(1:100) |
| Antibody | anti-EcR-B1 (Mouse monoclonal) | Developmental Studies Hybridoma Bank | Cat# AB_2154902, RRID:AB_2154902 | IF(1:50) |
| Antibody | anti-Trio (Mouse monoclonal) | Developmental Studies Hybridoma Bank | Cat# AB_528494, RRID:AB_528494 | IF(1:50) |
| Antibody | anti-CD8 (Rat monoclonal) | Thermo Fisher Scientific | Cat# MCD0800, RRID:AB_10392843 | IF(1:100) |

*Continued on next page*

*Continued*

| Reagent type (species) or resource | Designation | Source or reference | Identifiers | Additional information |
|---|---|---|---|---|
| Antibody | anti-HA (Rat monoclonal) | Roche | Cat# 11867423001, RRID:AB_390918 | IF(1:100) |
| Antibody | anti-GFP (Rabbit polyclonal) | Thermo Fisher Scientific | Cat#: A-11122; RRID:AB_221569 | IF(1:750) |
| Antibody | anti-Chinmo (Guinea pig polyclonal) | Sokol Lab; ref #46 | Cat#: A-11122; RRID:AB_221569 | IF(1:750) |
| Antibody | anti-rabbit Alexa 488 (Goat polyclonal) | Thermo Fisher Scientific | Cat# A-11034, RRID:AB_2576217 | IF(1:750) |
| Antibody | anti-rat Alexa 546 (Goat polyclonal) | Thermo Fisher Scientific | Cat# A-11081, RRID:AB_25335867 | IF(1:750) |
| Antibody | anti-guinea pig Alexa 647 (Goat polyclonal) | Thermo Fisher Scientific | Cat# A-21450, RRID:AB_2534125 | IF(1:750) |
| Antibody | anti-mouse Alexa 647 (Goat polyclonal) | Jackson ImmunoResearch lab, Inc | Cat# 115-605-166, RRID:AB_2338914 | IF(1:750) |
| Chemical compound, drug | Formaldehyde 37% solution | Sigma-Aldrich | Cat# 252549 | 4% |
| Chemical compound, drug | Paraformaldehyde 16% solution | Electron Microscopy Sciences | Cat# 15710 | 4% |
| Chemical compound, drug | SlowFade Gold Antifade Mountant | Thermo Fisher Scientific | Cat# S36936 | Anti-quenching |
| Software, algorithm | LSM | Zeiss | n.a. | Image processing |
| Software, algorithm | Photoshop CS6 | Adobe | n.a. | Image processing |
| Software, algorithm | Activity monitor system | DroBot, Inc | n.a. | Behavioral analysis |

## Experimental model and subject details

Flies were cultured in a room maintained at 25°C (±1.5°C) and 50–65% humidity for all experiments. For most experiments, flies were used with no selection for sex; therefore, roughly equal numbers of males and females were used. However, since homozygous *Ca-α1T-GFSTF* females and hemizygous *Ca-α1T-GFSTF* males are relatively feeble, heterozygous Ca-α1T-GFSTF females were selected for the analysis in *Figures 1G, H, 2A, B, 3A, B, and 4I, J*, *Figure 2—figure supplement 2*, *Figure 4—figure supplements 3 and 4* due to the cytolocation of the *Ca-α1T* gene on the X chromosome. In addition, males were selected for the behavioral assay to mitigate complications of egg laying behaviors on locomotion tracking.

## Fly strains

The fly strains used in this study were as follows. Most strains are available from either Bloomington *Drosophila* stock center (BDSC), Kyoto *Drosophila* stock center (DGGR), or Vienna *Drosophila* stock center (VDRC). (1) *Ab-GFP* (BDSC38626); (2) *Lac-FSVS* (DGGR115308); (3) *E93-GFSTF* (BDSC59412); (4) *Ca-α1T-GFSTF* (BDSC61800); (5) *hs-FLP[12],UAS-mCD8::GFP* (BDSC28832); (6) *tubP-GAL80,FRT[40A]* (BDSC5192); (7) *chinmo[1],FRT[40A]* (BDSC59969); (8) *GAL4-OK107* (BDSC854); (9) *UAS-mCD8::RFP [attp40]* (BDSC32219); (10) *UAS-mCD8::RFP [attp2]* (BDSC32218); (11) *UAS-E93 RNAi* (BDSC57868); (12) *UAS-E93 RNAi* (VDRC104390); (13) *UAS-Ca-α1T RNAi* (BDSC39029); (14) *GAL4-c739* (BDSC7362); (15) *44E04-LexA::P65* (BDSC52736); (16) *70F05-LexA::P65* (BDSC523629); (17) *lexAop2-myr::GFP* (*Pfeiffer et al., 2008*); (18) *lexAop2-mCD8:: RFP* (*Pfeiffer et al., 2008*); (19) *FRT[82B],tubP-GAL80* (BDSC5135); (20) *FRT[82B]* (BDSC86313); (21) *E93[Δ11]* (BDSC93128); (22) *E93(EP)* (BDSC30179); (23) *UAS-E93-A [VK37]* (this study); (24) *UAS-E93-B [VK37]* (this study); (25) *GAL4-201Y* (BDSC4440); (26) *mamo[H/l]-HA* (*Chu et al., 2024*); (27) *mamoD [D-G]-HA* (*Chu et al., 2024*); (28) *UAS-chinmo RNAi [VK37]* (*Lai et al., 2022*); (29) *UAS-LUC-let7* (BDSC41171); (30) *UAS-Ssyp RNAi* (VDRC33011); (31) *UAS-mamo RNAi* (BDSC44103); (32) *UAS-ab* (BDSC23639); (33) *UAS-ab-HA* (FlyORF000705); (34) *UAS-Dcr2* (BDSC24651); (35) *UAS-ab RNAi* (VDRC104582); (36) *13F02-p65.AD* (BDSC68291); (37) *70F05-GAL4.DBD* (BDSC69380). The *UAS-E93-A* and *UAS-E93-B* transgenes were generated using standard molecular biology methods to clone cDNA fragments derived from fully sequenced EST clones, GH10557 and LP08695 (available from *Drosophila* Genomics Resource Center), carrying *E93-A*

and *E93-B* isoforms into the attB-UAST vector. The generation of *UAS-E93-A* and *UAS-E93-B* transgenes and fly strains was performed by WellGenetics, Inc.

## RNAi knockdown and overexpression experiments and MARCM clonal analyses

*UAS-RNAi* and *UAS-transgene* lines were crossed to GAL4-107 and GAL4-201Y for knockdown and overexpression of genes of interest in KCs. Mosaic clones for the MARCM studies were generated as previously described (*Lee and Luo, 1999*). In short, mosaic clones of *chinmo*[1] and *E93*[Δ11] mutations were induced by 35 min of heat shock using *hs-FLP*[12] in newly hatched larva. Dissection, immunostaining, and mounting of adult brains were performed as described in a standard protocol (*Lee and Luo, 1999*). Primary antibodies used in this study included guinea pig antibody against Chinmo (1:1000, Sokol laboratory *Chawla et al., 2016*), rat monoclonal antibody against mCD8 (1:100, Thermo Fisher Scientific), rabbit antibody against GFP (1:750, Thermo Fisher Scientific), and mouse monoclonal antibodies against EcR-B1 (1:50, DSHB), Fas2 (1:100, DSHB) and Trio (1:50, DSHB). Secondary antibodies conjugated to different fluorophores (Alexa 488, Alexa 546, and Alexa 647; Thermo Fisher Scientific and Jackson ImmunoResearch Lab, Inc) were used at 1:750 dilutions. Immunofluorescence images were collected by confocal microscopy on a Zeiss LSM 700, projected using the LSM browser and processed in Adobe Photoshop CS6. All data are representative of more than 3 brains per genotype.

## Distance measurement in the behavioral assay and statistical analysis

Individual fly activities were recorded and analyzed using the activity monitor system developed by DroBot, Inc. This system used an open-resourced software pySolo to track individual moving flies (*Gilestro and Cirelli, 2009*). Average speed and standard deviation of individual flies were calculated according to total traveling distance within 30 min periods from the first day to the fifth day, as shown in *Figure 2I*, *Figure 2—figure supplement 5E*. Student's *t*-test was used for statistical analysis to compare datasets with two groups in *Figure 2—figure supplement 4*.

## Acknowledgements

We thank the TRiP at Harvard Medical School (NIH/NIGMS R01-GM084947) for providing the transgenic RNAi fly stocks used in this study. We also thank DroBot Inc for designing the behavioral assay system used in this study. This work was supported by the National Science and Technology Council (NSTC-112-2311-B-001-029) and Thematic Research Program of Academia Sinica (AS-TP-113-L01), Taiwan.

## Additional information

### Funding

| Funder | Grant reference number | Author |
|---|---|---|
| National Science and Technology Council | NSTC-112-2311-B-001-029 | Hung-Hsiang Yu |
| Academia Sinica | AS-TP-113-L01 | Hung-Hsiang Yu |

The funders had no role in study design, data collection, and interpretation, or the decision to submit the work for publication.

### Author contributions

Pei-Chi Chung, Conceptualization, Data curation, Formal analysis, Validation, Investigation, Visualization, Methodology, Writing – original draft; Kai-Yuan Ku, Sao-Yu Chu, Methodology; Chen Chen, Formal analysis, Visualization, Methodology; Hung-Hsiang Yu, Conceptualization, Data curation, Formal analysis, Supervision, Funding acquisition, Validation, Investigation, Visualization, Methodology, Writing – original draft, Project administration, Writing – review and editing

## Author ORCIDs
Pei-Chi Chung (ID) https://orcid.org/0000-0001-8081-0967
Kai-Yuan Ku (ID) https://orcid.org/0009-0000-5299-2249
Sao-Yu Chu (ID) https://orcid.org/0000-0002-5295-1663
Hung-Hsiang Yu (ID) https://orcid.org/0000-0002-2430-9251

Reviewer #1 (Public review): https://doi.org/10.7554/eLife.108173.3.sa1
Reviewer #2 (Public review): https://doi.org/10.7554/eLife.108173.3.sa2
Author response https://doi.org/10.7554/eLife.108173.3.sa3

---

# Additional files

## Supplementary files
MDAR checklist

Supplementary file 1. Expression patterns of GFP lines in *Figure 1—figure supplement 1*.

Supplementary file 2. Genotypes of flies shown in each figure panel.

## Data availability
Dataset uploaded to Dryad at: https://doi.org/10.5061/dryad.7wm37pw7n.

The following dataset was generated:

| Author(s) | Year | Dataset title | Dataset URL | Database and Identifier |
|---|---|---|---|---|
| Yu HH | 2026 | Genetic network shaping Kenyon cell identity and function in *Drosophila* mushroom bodies | https://doi.org/10.5061/dryad.7wm37pw7n | Dryad Digital Repository, 10.5061/dryad.7wm37pw7n |

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
