## [Editor Report · eLife Assessment]

This **fundamental** study uses the Drosophila mushroom body as a model to understand the molecular machinery that controls the temporal specification of neuronal cell types. With **convincing** experimental evidence, the authors make the finding that the Pipsqueak domain-containing transcription factor Eip93F plays a central role in specifying a later-born neuronal subtype while repressing gene expression programs for earlier subtypes.

---

## [Referee Report · Reviewer #1 (Public review)]

Summary:

The temporal regulation of neuronal specification and its molecular mechanisms are important problems in developmental neurobiology. This study focuses on Kenyon cells (KCs), which form the mushroom body in *Drosophila melanogaster*, in order to address this issue. Building on previous findings, the authors examine the role of the transcription factor Eip93F in the development of late-born KCs. The authors revealed that Eip93F controls the activity of flies at night through the expression of the calcium channel Ca-α1T. Thus, the study clarifies the molecular machinery that controls temporal neuronal specification and animal behavior.

Strengths:

The convincing results are based on state-of-the-art molecular genetics, imaging, and behavioral analysis.

---

## [Referee Report · Reviewer #2 (Public review)]

Summary:

Understanding the mechanisms of neural specification is a central question in neurobiology. In Drosophila, the mushroom body (MB), which is the associative learning region in the brain, consists of three major cell types: γ, α'/β' and α/β kenyon cells. These classes can be further subdivided into seven subtypes, together comprising ~2000 KCs per hemi-brain. Remarkably, all of these neurons are derived from just four neuroblasts in each hemisphere. Therefore, a lot of endeavours are put to understand how the neuron is specified in the fly MB.

Over the past decade, studies have revealed that MB neuroblasts employ a temporal patterning mechanism, producing distinct neuronal types at different developmental stages. Temporal identity is conveyed through transcription factor expression in KCs. High levels of Chinmo, a BTB-zinc finger transcription factor, promote γ-cell fate (Zhu et al., Cell, 2006). Reduced Chinmo levels trigger expression of mamo, a zinc finger transcription factor that specifies α'/β' identity (Liu et al., eLife, 2019). However, the specification of α/β neurons remains poorly understood. Some evidence suggests that microRNAs regulate the transition from α'/β' to α/β fate (Wu et al., Dev Cell, 2012; Kucherenko et al., EMBO J, 2012). One hypothesis even proposes that α/β represents a "default" state of MB neurons, which could explain the difficulty in identifying dedicated regulators.

The study by Chung et al. challenges this hypothesis. By leveraging previously published RNA-seq datasets (Shih et al., G3, 2019), they systematically screened BAC transgenic lines to selectively label MB subtypes. Using these tools, they analyzed the consequences of manipulating E93 expression and found that E93 is required for α/β specification. Furthermore, loss of E93 impairs MB-dependent behaviors, highlighting its functional importance.

Strengths:

The authors conducted a thorough analysis of E93 manipulation phenotypes using LexA tools generated from the Janelia Farm and Bloomington collections. They demonstrated that E93 knockdown reduces expression of Ca-α1T, a calcium channel gene identified as an α/β marker. Supporting this conclusion, one LexA line driven by a DNA fragment near EcR (R44E04) showed consistent results. Conversely, overexpression of E93 in γ and α'/β' Kenyon cells led to downregulation of their respective subtype markers.

Another notable strength is the authors' effort to dissect the genetic epistasis between E93 and previously known regulators. Through MARCM and reporter analyses, they showed that Chinmo and Mamo suppress E93, while E93 itself suppresses mamo. This work establishes a compelling molecular model for the regulatory network underlying MB cell-type specification.

Weaknesses:

The interpretation of E93's role in neuronal specification requires caution. Typically, two criteria are used to establish whether a gene directs neuronal identity:

(1) gene manipulation shifts the neuronal transcriptome from one subtype to another, and

(2) gene manipulation alters axonal projection patterns.

The results presented here only partially satisfy the first criterion. Although markers are affected, it remains possible that the reporter lines and subtype markers used are direct transcriptional targets of E93 in α/β neurons, rather than reflecting broader fate changes. Future studies using transcriptomics would provide a more comprehensive assessment of neuronal identity following E93 perturbation.

With respect to the second criterion, the evidence is also incomplete. While reporter patterns were altered, the overall morphology of the α/β lobes appeared largely intact after E93 knockdown. Overexpression of E93 in γ neurons produced a small subset of cells with α/β-like projections, but this effect warrants deeper characterization before firm conclusions can be drawn.

Overall, this study has nicely shown that E93 can regulate α/β neural identities. Further studies on the regulatory network will help to better understand the mechanism of neurogenesis in mushroom body.

---

## [Author Response]

The following is the authors’ response to the original reviews.

**Public Reviews:**

**Reviewer #1 (Public review):**
Summary:The temporal regulation of neuronal specification and its molecular mechanisms are important problems in developmental neurobiology. This study focuses on Kenyon cells (KCs), which form the mushroom body in *Drosophila melanogaster*, in order to address this issue. Building on previous findings, the authors examine the role of the transcription factor Eip93F in the development of late-born KCs. The authors revealed that Eip93F controls the activity of flies at night through the expression of the calcium channel Ca-α1T. Thus, the study clarifies the molecular machinery that controls temporal neuronal specification and animal behavior.Strengths:The convincing results are based on state-of-the-art molecular genetics, imaging, and behavioral analysis.Weaknesses:Temporal mechanisms of neuronal specification are found in many nervous systems. However, the relationship between the temporal mechanisms identified in this study and those in other systems remains unclear.

We have discussed the temporal mechanisms between different nervous systems at the beginning of the Discussion section.

**Reviewer #2 (Public review):**
Summary:Understanding the mechanisms of neural specification is a central question in neurobiology. In Drosophila, the mushroom body (MB), which is the associative learning region in the brain, consists of three major cell types: γ, α'/β', and α/β kenyon cells. These classes can be further subdivided into seven subtypes, together comprising ~2000 KCs per hemi-brain. Remarkably, all of these neurons are derived from just four neuroblasts in each hemisphere. Therefore, a lot of endeavors are put into understanding how the neuron is specified in the fly MB.Over the past decade, studies have revealed that MB neuroblasts employ a temporal patterning mechanism, producing distinct neuronal types at different developmental stages. Temporal identity is conveyed through transcription factor expression in KCs. High levels of Chinmo, a BTB-zinc finger transcription factor, promote γ-cell fate (Zhu et al., Cell, 2006). Reduced Chinmo levels trigger expression of mamo, a zinc finger transcription factor that specifies α'/β' identity (Liu et al., eLife, 2019). However, the specification of α/β neurons remains poorly understood. Some evidence suggests that microRNAs regulate the transition from α'/β' to α/β fate (Wu et al., Dev Cell, 2012; Kucherenko et al., EMBO J, 2012). One hypothesis even proposes that α/β represents a "default" state of MB neurons, which could explain the difficulty in identifying dedicated regulators.The study by Chung et al. challenges this hypothesis. By leveraging previously published RNA-seq datasets (Shih et al., G3, 2019), they systematically screened BAC transgenic lines to selectively label MB subtypes. Using these tools, they analyzed the consequences of manipulating E93 expression and found that E93 is required for α/β specification. Furthermore, loss of E93 impairs MB-dependent behaviors, highlighting its functional importance.Strengths:The authors conducted a thorough analysis of E93 manipulation phenotypes using LexA tools generated from the Janelia Farm and Bloomington collections. They demonstrated that E93 knockdown reduces expression of Ca-α1T, a calcium channel gene identified as an α/β marker. Supporting this conclusion, one LexA line driven by a DNA fragment near EcR (R44E04) showed consistent results. Conversely, overexpression of E93 in γ and α'/β' Kenyon cells led to downregulation of their respective subtype markers.Another notable strength is the authors' effort to dissect the genetic epistasis between E93 and previously known regulators. Through MARCM and reporter analyses, they showed that Chinmo and Mamo suppress E93, while E93 itself suppresses Mamo. This work establishes a compelling molecular model for the regulatory network underlying MB cell-type specification.Weaknesses:The interpretation of E93's role in neuronal specification requires caution. Typically, two criteria are used to establish whether a gene directs neuronal identity:(1) gene manipulation shifts the neuronal transcriptome from one subtype to another, and(2) gene manipulation alters axonal projection patterns.The results presented here only partially satisfy the first criterion. Although markers are affected, it remains possible that the reporter lines and subtype markers used are direct transcriptional targets of E93 in α/β neurons, rather than reflecting broader fate changes. Future studies using single-cell transcriptomics would provide a more comprehensive assessment of neuronal identity following E93 perturbation.

We do plan conduct multi-omics experiments to provide a more comprehensive assessment of neuronal identity upon loss-of-function of E93. However, omics results take time to be conducted and analyzed, so the result will be summarized in a future manuscript.

With respect to the second criterion, the evidence is also incomplete. While reporter patterns were altered, the overall morphology of the α/β lobes appeared largely intact after E93 knockdown. Overexpression of E93 in γ neurons produced a small subset of cells with α/β-like projections, but this effect warrants deeper characterization before firm conclusions can be drawn. While the results might be an intrinsic nature of KC types in flies, the interpretation of the reader of the data should be more careful, and the authors should also mention this in their main text.

We have toned down our description on the effect of E93 (especially in the loss-offunction) in specifying the α/β-specific cell identity and discussed whether unidentified regulators would work together with E93 in α/β neural fate specification.

**Recommendations for the authors:**

**Reviewer #1 (Recommendations for the authors):**
(1) Changes in nighttime activity in flies upon knocking down Ca_α1T and Eip93F are interesting (Fig. 2C). However, examining the morphological changes in the mushroom body under these conditions would be essential.

We did not find the morphological change of mushroom body lobes by examining with the Fas2 staining (shown in Figure S8D).

(2) Temporal mechanisms of neuronal specification have been identified in various nervous systems, including the embryonic central nervous system (CNS), the optic lobe of Drosophila, and the nervous systems of other organisms. The Discussion section should address the relationship between the temporal mechanisms identified in this study and those identified in other systems.

We have discussed the temporal mechanisms between different nervous systems at the beginning of the Discussion section.

(3) Eip93F is an Ecdysone-induced protein. In the Discussion section, the authors should discuss the relationship between the ecdysone signal and the roles of Eip93F.

We have added the discussion on the relationship between the ecdysone signal and the roles of Eip93F.

**Reviewer #2 (Recommendations for the authors):**
(1) The behavioral effect of Ca-α1T knockdown is pretty interesting. But how the downregulation of Ca-α1T in the mushroom body can affect locomotion is puzzling. Even though the mushroom body is known to suppress locomotion (Matin et al., Learn Mem, 1998), the real results are opposite. Can authors give further explanation in the discussion? Also, the behavioral experiments are hard to interpret, given that Figure 2C(1) and Figure 2C(3) as a control, also vary a lot. Since the behavioral experiments don't affect the main conclusion of the paper, I would suggest removing that part or adding more explanation in the discussion.

First, we have discussed the puzzling part on the MB influence in locomotion between the previous study using tetanus toxin light chain (TeNT-Ln) and our Ca-α1T knockdown result. It is possible that the different effect is derived from TeNT-Ln’s function in MB axons and Ca-α1T’s function in MB dendrites. Secondly, we have re-conducted the behavioral results using a new α/β driver (13F02-AD/70F05-DBD) to replace our initial behavioral results (using c739-GAL4, which would cause the abnormal wing when drives E93 RNAi expression; see S8C(2) Fig). Current results (now in Fig 2I) are more consistent in control groups.

(2) In the manuscript, the authors use "subtype" to describe γKC, α'/β'KC and α/βKC in the fly MB. However, in most of the literature, people use "main types" to summarize these three types, and "subtype" is mostly about the difference in γd, γm, α'/β'ap, α'/β'm, α/βp, α/βs and α/βc KC (Shih et al., G3, 2019). Replacing "subtypes" with "main types" will help to increase the clarity.

We have replaced "KC subtypes" with "main KC types" or just “KC types”.

(3) The authors have identified a lot of new markers for the KC cell types, and some of them are used in this manuscript. It will be helpful if they can have a figure to summarize the markers they used in this study and what cell types they labeled.

We have summarized expression patterns of these markers in Supplemental table 1.

(4) In the method, the authors mentioned that only females were selected for analysis of Ca-α1T-GFSTF. Could the authors explain the reasons in more detail?

Since homozygous *Ca-α1T-GFSTF* female flies and hemizygous *Ca-α1T-GFSTF* male are a bit sick and hard to collect, we therefore used heterozygous *Ca-α1T-GFSTF* female in our experiments. I have added this description in the Materials and Methods section.

(5) Figure S1: The legend of magenta fluorescence is missing. Please add which protein is shown in magenta.

We have added the legend of magenta fluorescence, which is Trio.

(6) The detailed genotypes of Figure 2C and Figure S7 are missing in Supplementary Table 1. Please include that, so that readers can know the genetic background.

We have added genotypes of Figure 2I (previously Figure 2C) and Figure S8 (previously as Figure S7) in Supplementary Table 2.

(7) Figure 2D-G: It will be helpful if the authors can outline the lobe (γ, α'/β', and α/β) in the figure, which will help readers to understand the images.

We have outlined α, α', β, β' and γ lobes in Figure 2C-F (previously as Figure 2D-G).